# Unlocking the Potential of mHealth: Integrating Behaviour Change Techniques in Hypertension App Design

**DOI:** 10.3390/ijerph22101487

**Published:** 2025-09-25

**Authors:** Emily Motta-Yanac, Riley Victoria, Naomi J. Ellis, Christopher James Gidlow

**Affiliations:** 1Centre for Health and Development (CHAD), Staffordshire University, Stoke-on-Trent ST4 2DF, UK; victoria.riley@staffs.ac.uk (R.V.); n.j.ellis@staffs.ac.uk (N.J.E.); 2School of Medicine, Keele University, University Road, Keele ST5 5BG, UK; c.gidlow@keele.ac.uk; 3Research and Innovation Department, Midlands Partnership University NHS Foundation Trust, St Georges Hospital, Corporation Street, Stafford ST16 3AG, UK

**Keywords:** mHealth, behaviour change, hypertension

## Abstract

**Background:** Smartphone apps offer a promising avenue for delivering scalable interventions for hypertension self-management. This study aimed to characterise the behaviour change technique ontology (BCTO) elements present in apps available on popular platforms, map the theoretical domains framework (TDF), and describe the apps’ functionalities. **Methods:** A comprehensive search of app stores was conducted to identify relevant hypertension self-management apps. The identified apps were then analysed for BCTO elements, which were subsequently mapped to TDF. App functionalities and quality were assessed as well. **Results:** Functionalities such as self-monitoring were consistently observed across all app types, aligning with established hypertension self-management strategies. However, other key functionalities, including goal setting via reminders, communication with healthcare professionals, and data export capabilities, were less prevalent. AI-empowered apps incorporated a broader range of behaviour change techniques compared to non-AI and RCT-tested apps, suggesting a potentially more comprehensive approach to supporting behaviour change. The domains of “Knowledge”, “Emotions”, “Behavioural regulation”, “Skills”, and “Beliefs about Consequences” were most frequently targeted by app developers. AI-empowered apps incorporated a broader range of BCTs compared to non-AI and RCT-tested apps, potentially offering more comprehensive support for behaviour change. **Conclusions:** While existing hypertension self-management apps incorporate a variety of BCTs, there is room for improvement in terms of incorporating a wider range of functionalities and BCTOs, particularly those targeting more intrinsic and habitual aspects of behaviour.

## 1. Introduction

Mobile health (mHealth) apps have become relatively prevalent in mobile phone platforms [1]. These digital health interventions (DHIs) have an impact on the delivery of health and wellness, offering scalable solutions from fitness and general wellness to mental health support and chronic disease management [2,3]. The digital health landscape is rapidly evolving, with a growing volume and variety of mHealth apps, with over 350,000 apps available as of 2020 [2]. These apps play a crucial role in fostering health literacy and improving access to care, particularly in underserved regions, thereby reshaping how individuals engage with their health and interact with healthcare systems [4]. This widespread adoption positions mHealth apps as critical interventions, particularly for chronic conditions such as hypertension, which demand sustained self-management strategies and continuous support.

Hypertension, a global health concern, poses a public health challenge due to its high prevalence and severe impact on health outcomes. Globally, it is estimated that around 1.3 billion adults have high blood pressure (BP) [5], with 1.2 billion failing to achieve controlled blood pressure levels [6]. Uncontrolled hypertension is the leading preventable risk factor for cardiovascular disease and premature death worldwide [6]. Effective management of hypertension focuses on controlling BP; therefore, interventions usually cover promoting healthy habits, giving adherence feedback to patients, self-monitoring of BP, using pill boxes and other special packaging, and motivational interviewing [7]. These components are linked to behavioural change; however, this approach in health interventions is challenging as behavioural theories are usually overlooked. Behavioural psychology professionals have developed several theories that try to explain how different factors are linked to behaviour. Commonly used theories in health-related interventions include the Capability, Opportunity, Motivation, Behaviour Model [8].

Many mHealth apps are targeted at supporting people with self-managing hypertension by offering self-monitoring activities, reminders, tailored information, and feedback [9,10]. The rapid growth of the use of these apps has been accompanied by novel technology advances, such as artificial intelligence (AI). This approach offers a promising avenue for supporting behaviour change, but its insufficient evaluation hinges on incorporating evidence-based behaviour change techniques (BCTs).

Given these challenges and the growing roles of mHealth, this study aims to answer key questions regarding Behaviour Change Technique Ontology (BCTO) elements present in mobile phone apps designed for hypertension self-management; how the functionalities, BCTOs, and theoretical domains employed in commercially available apps compare to those in research-based apps; and how these insights can elucidate the underlying mechanism of action driving behavioural change interventions in hypertension apps.

This study aims to identify BCTs that influence hypertension management interventions delivered by a smartphone app using the Behaviour Change Technique Ontology (BCTO) taxonomy. Moreover, it aims to investigate and compare the functionality, BCTOs, and theoretical domains employed in commercially available apps and research-based apps for hypertension self-management to elucidate the underlying mechanisms of action driving these behavioural change interventions.

## 2. Materials and Methods

This study used a multi-faceted, systematic approach to identify, code, and evaluate mobile apps (AI-empowered, RCT-tested, and non-AI apps) targeting hypertension self-management. We employed taxonomic coding to categorise the various features and functionalities offered by each app. Then, we used narrative synthesis to integrate and interpret the findings from individual app assessments, providing a holistic overview of the current landscape.

The search for AI-empowered apps was undertaken in app stores of two types of smartphones in the United Kingdom—iPhone (Apple Stores) and Android (Google Play), which served as a primary source of publicly available apps and associated descriptive information (considered grey literature). Non-AI (NAI) apps were drawn from a systematic review study on smartphone apps for hypertension self-management [11]. Other retrieved NAI apps were RCT-tested from a meta-analysis study [12]. All apps included were made for English-speaking users and have a minimum of a 3.0 user rating.

Taxonomic coding was employed to systematically categorise the various features and functionalities present in each app. The BCTO [13] was used to analyse the intervention components, providing a structured understanding of how these apps aim to influence user behaviour, followed by a linking between the identified BCTO and the Theoretical Domains Framework (TDF) [14]. We relied on a previously validated expert consensus framework that maps BCTs to TDF domains for health interventions [15].

Additionally, we conducted a quality assessment using the Mobile Application Rating Scale (MARS) [16], which comprises four primary dimensions: “engagement,” “functionality,” “aesthetics,” and “information.” Each item was rated using a 5-point Likert scale (1 = Inadequate, 2 = Poor, 3 = Acceptable, 4 = Good, 5 = Excellent). RCT-tested apps were excluded from the analysis as insufficient information was available to complete the assessment.

Descriptive statistics were used to summarise app features, BCTs, and quality scores, while comparative analyses were conducted to identify differences in BCT density, TDF targeting, and quality metrics between AI-empowered, RCT-tested, and non-AI (NAI) apps.

## 3. Results

### 3.1. Identification of Apps

Figure 1 summarises the search results of the apps included. This study considered a total of 24 apps across all app types (NAI, AI-empowered, and RCT-tested).

The search yielded 33 AI-based apps in the two app platforms (30 in the Android Google Play Store and 3 in the iPad Apple App Store). Of these, seven apps met the inclusion criteria. For the NAI and RCT-tested apps, 10 and 9 apps met the inclusion criteria, respectively.

### 3.2. App Characteristics

Most of the included apps (8, 47%) were available to download on both Android and Apple operating systems. The remaining apps were available only through Apple (6, 35%) and Android operating systems (3, 18%). The reviewed apps’ version dates ranged from 2020 to 2024. According to the number of downloads, half of the included Android apps (5/11) had over 100,000 downloads. Information on the number of downloads was not available on Apple apps. User ratings were available in most selected apps (12/17) and ranged from 3.8 to 5.0. An extended description is shown in Appendix A.

### 3.3. Functionalities of the Apps

The main functions of the apps were categorised as self-monitoring ability, goal setting, reminders, educational information, feedback, stress management, communication with HPCs and others, export of user’s data to others via email, and prevention. All apps presented at least one of the functionalities, regardless of app type (Table 1).

Self-monitoring was the most common functionality in the apps, with 24 apps (100%) offering this functionality. These apps uniformly provided self-monitoring capabilities for tracking blood pressure readings through various formats, such as graphs and tables. Most of them (79%, 19/24) required a wireless monitor connected to the app. A few apps (3/24, 12.5%) enabled users to track medications, along with other apps tracking physical activity (3/24, 12.5%), weight (5/24, 20.8%), mood (2/24, 8.3%), and dietary (3/24, 12.5%). Most NAI apps monitored more than one health data (5/10, 50%). Only one RCT-tested app monitored blood sugar along with BP [17]. The second most common functionality was providing automatic feedback (19/24, 79%) after loading each BP reading into the app.

Automatic reminders and alert components prompt self-monitoring by reminding patients about their medication time, BP measurements, hospital visits or personal goals, and engagement with the app, a feature included in 18/24 (75%) of the apps. Other frequent components were exporting their entered data over time directly to others (14/24, 58.3%), enabling users to communicate with HCPs or others, such as family, friends or other users (11/24, 45.8%), providing educational information (9/24, 37.5%) for risks of uncontrolled BP and benefits of controlled BP, and enabling goal setting (8/24, 33.3%) of BP, weight and physical activity. The mentioned functions had a similar frequency between RCT-tested and AI-empowered apps. Stress management and prevention features were uniquely found in AI-empowered apps. Prevention functions consisted of preventing high BP readings by predicting BP levels and identifying health risk indicators (weight, stress, physical activity, salt intake).

### 3.4. Quality Assessment Using MARS

The MARS was used to evaluate the quality of AI-empowered and NAI apps (N = 17). The overall mean MARS quality score was 4.10 (SD 0.25), indicating good quality (Appendix A). MARS scores for each dimension by type of app are shown in Figure 2. All apps had at least an appropriate quality rating (>3), with 71% of AI-empowered and 50% of NAI-based apps classified as having an excellent MARS score.

Functionality scored highest among the four objective MARS dimensions for both app types (AI-empowered: 4.71, SD 0.44; NAI: 4.58, SD 0.75), followed by aesthetics (AI-based: 4.43, SD 0.60; NAI: 4.07, SD 0.73), information (AI-empowered: 4.10, SD 0.52; NAI: 3.86), and engagement (AI-empowered: 3.66, SD 0.56; NAI: 3.56). While the mean values for each MARS dimension were generally lower for NAI apps compared to AI-based apps, the differences were not statistically significant.

### 3.5. BCTOs and TDF Mapping

Across the 24 apps reviewed, a total of 25 unique BCTOs, as defined by the BCTO taxonomy, were identified, with an average of 6.5 BCTOs incorporated into each app. Figure 3 shows the frequency of BCTOs across the reviewed apps.

“Self-monitor behaviour” was present across all app types. Some BCTOs were exclusive to specific app types: “Arrange Support”, “substitute behaviour”, and “information about health consequences” were only found in AI-empowered apps. “Provide positive material consequence for behaviour” was unique to NAI apps. RCT-tested apps were the only group with “provide positive consequence for outcome of behaviour”, “directly restructure the social environment”, and “deliver emotional support”. Significant differences (*p* = 0.0152) were identified in the usage of BCTOs across the types of apps (Table 2). The findings suggest the potential of AI to create more impactful behaviour changes in mHealth interventions for hypertension self-management.

#### TDF Mapping

The BCTOs identified in the reviewed apps were linked to the TDF, with 11 TDF domains summarised based on whether the BCTOs appeared alone or in combination. Five domain combinations and four unique TDF domains were linked across the identified BCTOs. The most common single TDF mechanisms of action were “Skills” and “Environmental context and resources,” documented across all app types (Figure 4). The most frequent domain combinations across all types were “Knowledge/Behavioural regulation” and “Intentions and goals”.

Some patterns emerged in the combinations of TDF domains targeted by different app types. AI-empowered apps showed equal emphasis on “Knowledge/Emotions/Beliefs about consequences”, “Intention/Goals”, “Knowledge/Behavioural regulation”, and “Social influence/Emotions”, while the “Reinforcement/Beliefs about consequences/Social Influences” combination was absent. NAI apps frequently combined “Knowledge/Behaviour Regulation” but lacked the “Knowledge/Emotions/Beliefs about consequences” triad and used fewer TDF domains overall. RCT-tested apps, in contrast, exhibited a balanced distribution across all 11 domains, closely mirroring the profile of AI-empowerment apps, particularly in their focus on “Skills” and “Social influences/Emotions”.

## 4. Discussion

While this study focused on the design and functional elements in mHealth apps for hypertension management, it is crucial to acknowledge the broader evidence base concerning their effectiveness in improving health outcomes. Evidence has indicated that mHealth interventions for hypertension positively impact medication adherence, BP reduction, and overall self-management [18,19]. Functional components such as communication, reminders, education, monitoring, and feedback are often cited as effective in hypertension self-management programs [20]. However, demonstrating conclusive evidence of effectiveness remains a significant challenge due to methodological complexities, heterogeneity of interventions, and the rapid evolution of app features [21]. Despite these challenges, the consistent incorporation of evidence-based BCTs and high-quality design, as observed in some apps in our study, is considered a foundational element for developing mHealth solutions with the potential for greater clinical impact and user engagement [3,22]. Our findings thus provide valuable insights into the design characteristics that are likely to contribute to effective interventions, guiding future development and more targeted effectiveness research.

The study aimed to characterise the BCTOs across all types of apps available in the most used platforms designed for hypertension self-management, map these BCTOs to the TDF, and describe the apps’ functionalities. Functionalities that provide self-monitoring were a common feature across all app types, aligning with established self-management strategies for hypertension [23,24]. However, other functionalities, such as goal setting, communication with healthcare professionals, and data export capabilities, were less prevalent. Also, in this study, AI-empowered apps incorporate a broader range of BCTs compared to NAI and RCT-tested apps, potentially offering more comprehensive support for behaviour change. The MARS quality assessment revealed that AI-empowered apps generally scored higher in functionality, engagement, and information quality, indicating a superior user experience. The mapping of BCTOs to the TDF highlighted the varying mechanisms targeted by different app types, with AI-empowered apps exhibiting a more balanced approach across multiple domains, while NAI apps focused primarily on knowledge and behavioural regulation.

Delving deeper into our findings, the most common BCTOs revolved around self-monitoring behaviour and feedback on behaviour outcomes, which aligns with existing literature on effective self-management strategies for hypertension [11,25]. While every app incorporated at least one BCTO element, there were notable differences in the number and type of BCTOs across different app types. AI-empowered apps demonstrated a higher average and maximum number of BCTOs compared to other app types. These apps showed frequent goal-setting functions and exclusively enabled stress management functionalities, both widely recognised as key self-management strategies that can significantly benefit patients with hypertension [26]. Prior research indicates that techniques such as action planning were underrepresented, which aligns with our findings in the reviewed applications [27,28]. While our analysis quantified the presence and variety of BCTOs, this does not inherently guarantee effectiveness. The actual impact of an app is profoundly influenced by the accuracy and quality of implementation of these techniques, the overall user experience, and the potential for feature overload. An app’s effectiveness is also heavily dependent on its usability, intuitive design, and overall user experience, which drive user engagement and continued use [29]. Although our study’s MARS quality assessment provided insights into aspects like functionality and engagement, directly assessing the fidelity of BCT implementation or the precise impact of feature load on individual users was beyond its scope. Future research is needed to investigate these qualitative dimensions to fully understand how BCT presence translates into real-world health outcomes.

The mapping of BCTOs to TDF domains revealed “Skills” and “Environmental context and resources” were the most frequent single mechanisms, while “Knowledge/Behavioural regulation” and “Intentions and goals” were the most frequent combined mechanisms. This underscores the complexity of BCTs and the importance of considering both individual and combined mechanisms in app design. While ‘Knowledge’ and ‘Beliefs about Consequences’ were less frequently observed as individual targets, they were more commonly addressed in combination with other TDF domains, suggesting developers may recognise the interconnectedness of these domains. Evidence suggests that providing knowledge about hypertension may be more effective when combined with strategies to improve medication adherence or when users understand the potential consequences of uncontrolled blood pressure [30]. The absence of the “Knowledge/Emotions/Beliefs” triad in NAI-based apps might indicate a potential limitation in their ability to drive behaviour change compared to AI-based and RCT apps. This suggests the potential of AI and rigorous testing methodologies to create more impactful mHealth interventions, as also supported by literature on mental health digital interventions [31].

The integration of AI into mHealth apps, as observed in the broader range of BCTs and higher quality scores in our AI-empowered apps, presents significant opportunities for hypertension management, including enhanced personalisation of interventions, predictive analytics for risk assessment and treatment optimisation, real-time monitoring with actionable feedback, and improved processing of patient information [32,33,34]. However, these advancements are accompanied by substantial challenges. Ethical considerations such as transparency, bias, and patient autonomy, alongside critical issues of data security and privacy, pose considerable risks [35].

While this study characterised domains within hypertension apps, the regulatory and normative landscape that shapes their development and availability is broad. The regulation of mHealth apps is a complex and evolving area, with significant implications for patient safety, data privacy, and the validation of clinical claims [36]. In the European Union (EU), the Medical Device Regulation (MDR [EU]) 2017/745 classifies certain mHealth apps as medical devices, imposing strict requirements for clinical evaluation, quality management systems, and post-market surveillance [37]. However, the increased complexity of these regulations can also pose challenges for manufacturers, especially small- and medium-sized enterprises, potentially impacting the accessibility of innovative solutions in the European market. In contrast, the United States Food and Drug Administration (FDA) employs a risk-based approach, categorising mHealth apps into general wellness products, medical device data systems, and regulated medical devices, each with varying levels of oversight [38]. Globally, there is a recognised “regulatory gap” in digital health, where marketing authorisations do not always sufficiently signal the safety, efficacy, and ethical compliance of digital health technologies [39]. These evolving regulatory considerations underscore the importance of rigorous evaluation of mHealth apps, as performed in our study, to bridge the gap between technological innovation and validated clinical effectiveness. Our findings on the varying integration of BCTs and quality metrics across app types also highlight the need for regulatory frameworks to encourage the adoption of evidence-based components and user-centred design principles to ensure that apps deliver effective and safe interventions.

Beyond these technical and regulatory considerations, the successful clinical integration of mHealth apps demands overcoming significant practical barriers. These include ensuring seamless interoperability with existing electronic health records to facilitate secure data exchange [40], improving user engagement and digital literacy among patients [41], and addressing the need for adequate training and acceptance among healthcare providers, who often face systematic resistance and lack of familiarity with these tools [41].

This review presents some limitations. Primarily, all apps included, irrespective of their initial discovery method, were ultimately verified to be available in app stores and met the specified user rating criteria; the differential initial sourcing introduces a potential selection bias. AI-empowered apps were identified through direct searches in app stores, whereas NAI and RCT-tested apps were extracted from existing systematic reviews and meta-analyses. This distinction in how apps were initially found could lead to systematic differences in the characteristics of the cohorts. For example, apps prominently featured in academic reviews may tend to be older, more established, or specifically developed within research contexts, even if they are also publicly available. This methodological heterogeneity means that the observed differences in BCTs, functionalities, and quality metrics between app types might not solely reflect intrinsic technological distinctions (e.g., AI vs. non-AI) but could also be influenced by these varied discovery pathways.

## 5. Conclusions

This study investigated the extent to which the BCTO is reflected in the design of smartphone apps for hypertension self-management. Our analysis revealed that app developers predominantly targeted the Theoretical Domains Framework domains of “Knowledge”, “Emotions”, “Behavioural regulation”, “Skills” and “Beliefs about Consequences”. However, our findings also indicate opportunities to better incorporate a wider range of behaviour change techniques, particularly those targeting more intrinsic and habitual aspects of behaviour, such as “Goals” and “Social influences”. Furthermore, the assessment of app features and quality revealed considerable variation within the hypertension app landscape. While some apps demonstrated robust functionality and adherence to quality standards, others lacked key features or exhibited potential usability issues. This underscores the need for more standardised quality control measures in developing and disseminating mHealth apps for hypertension management. Overall, apps that effectively influence users’ feelings and habits related to hypertension management, while also adhering to quality standards and user-centred design principles, are likely to be more successful in promoting positive health outcomes.

## Figures and Tables

**Figure 1 ijerph-22-01487-f001:**
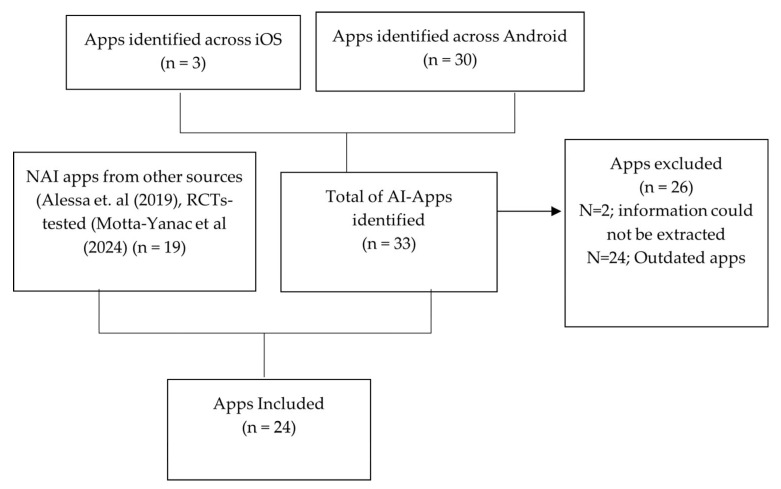
PRISMA flowchart of the screening results for the identification of the apps [11].

**Figure 2 ijerph-22-01487-f002:**
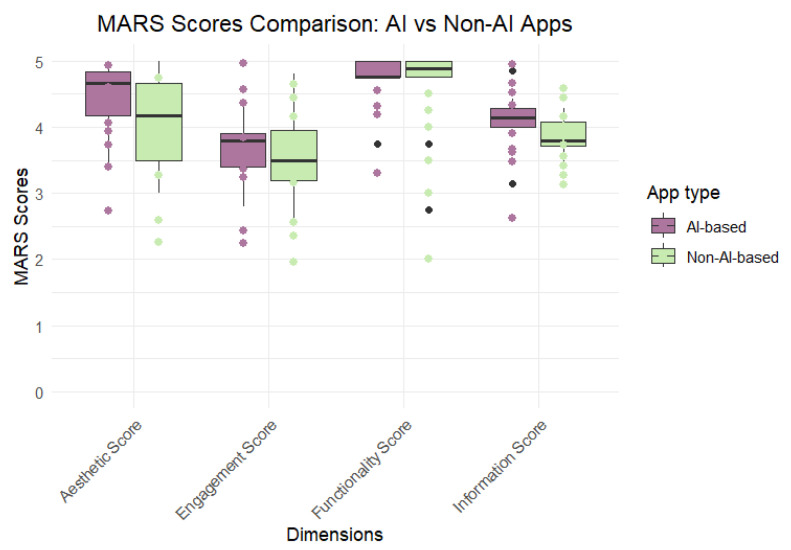
Mobile Application Rating Scale (MARS) scores by dimensions for each app type (AI-empowered and NAI). Black dots represent outliers for either group.

**Figure 3 ijerph-22-01487-f003:**
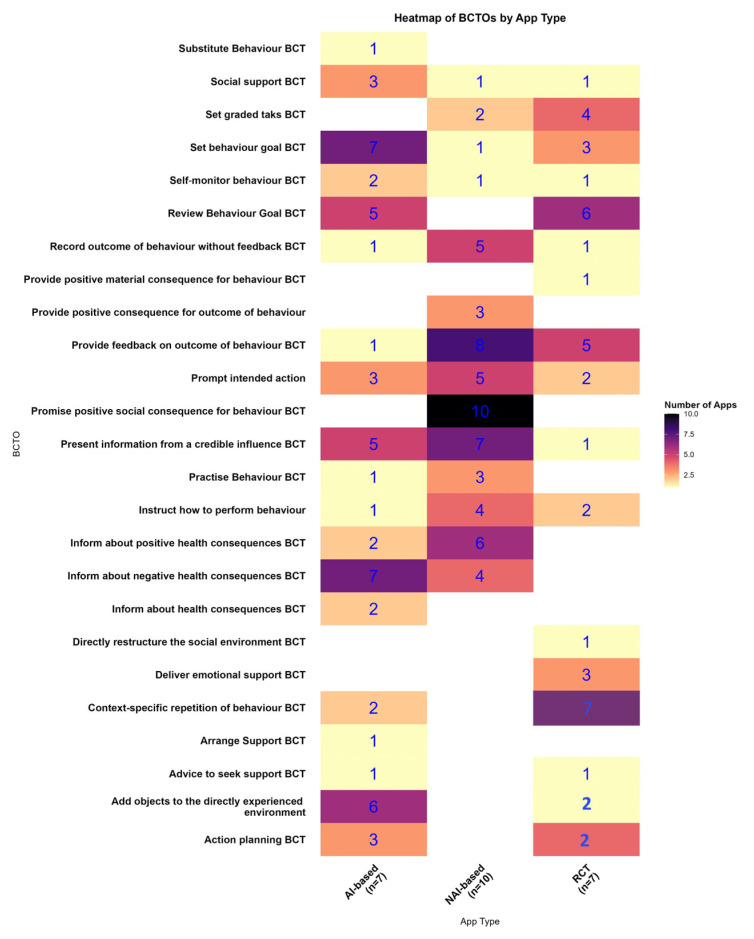
Visualisation of the distribution and frequencies of behaviour change technique ontologies (BCTOs) across the app type (AI-based, NAI-based and RCT app). RCT: randomised controlled trial; BCT: Behaviour change technique; NAI: non-AI.

**Figure 4 ijerph-22-01487-f004:**
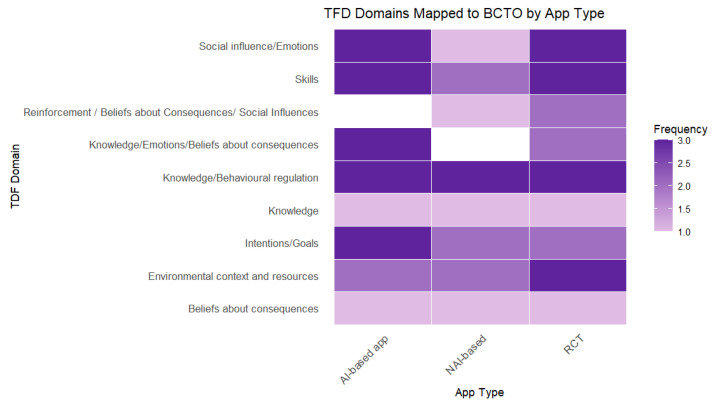
Heatmap of Theoretical Domains Framework (TDF) mapped into Behaviour Change Techniques Ontology (BCTO) across app types (AI, NAI and RCT-tested).

**Table 1 ijerph-22-01487-t001:** The frequency of functionalities across the selected apps.

Functionality	AI-Empowered (N = 7), n (%)	Non-AI (NAI) (N = 10), n(%)	RCT-Tested (N = 7), n (%)	Total (N = 24), n (%)
Self-Monitoring	7 (100)	10 (100)	7 (100)	24 (100)
Goal Setting	3 (42.8)	2 (20)	3 (42.8)	8 (33.3)
Reminders	5 (71.4)	9 (90)	4 (57.1)	18 (75)
Educational Information	3 (42.8)	2 (20)	4 (57.1)	9 (37.5)
Feedback	7 (100)	7 (70)	5 (71.4)	19 (79.2)
Stress management	2 (28.6)	0	0	2 (8.3)
Communication with HCPs and other	4 (57.1)	3 (30)	4 (57.1)	11 (45.8)
Export of user’s data to others via email	2 (28.6)	10 (100)	2 (28.6)	14 (58.3)
Prevention	6 (85.7)	0	0	6 (25)

RCT; Randomised Controlled Trial.

**Table 2 ijerph-22-01487-t002:** Descriptive statistics for the inclusion of BCTOs.

Total BCTOs		AI-Based (N = 7)	NAI-Based (N = 10)	RCTs (N = 7)	Total (N = 24)
25	Mean ± SD	7.57 ± 2.44	4.29 ± 2.66	2.45 ± 1.80	0.0152
	Median	7	4	2	
	Min, Max	4, 12	1, 10	1, 7	

## Data Availability

The original contributions presented in this study are included in the article/Appendix A. Further inquiries can be directed to the corresponding author.

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
