# Peer review of "Unlocking the Potential of mHealth: Integrating Behaviour Change Techniques in Hypertension App Design"

_ijerph, 2025, doi:10.3390/ijerph22101487_

Round 1

Reviewer 1 Report

Comments and Suggestions for Authors

This paper looks at an important topic, namely how smartphone apps, integrate behaviour change techniques (BCTs) to help patients manage high blood pressure on their own. The use of the Behaviour Change Technique Ontology (BCTO), the Theoretical Domains Framework (TDF) and the MARS assessment provides a solid methodological foundation. The results are relevant for both researchers and developers working at the interface between mHealth and the treatment of chronic diseases. Overall, the manuscript is well structured and provides new insights, particularly regarding the wider range of BCTs in AI-based apps. However, some areas could be improved to increase clarity, interpretability and impact.

The study evaluates the functions and design of the apps, but does not provide data on the effectiveness of the apps in improving health outcomes. While the exclusion of RCT-tested apps from MARS is understandable given the limited data available, a general discussion of the evidence base for effectiveness would strengthen the practical relevance of the findings.

The AI-based apps were selected directly from app stores, while NAI and RCT apps were sourced from existing reviews. This may introduce bias into the selection, as these app pools may differ systematically from one another. Please clarify whether this heterogeneity may have influenced the comparison and consider noting this as a limitation.

The analysis focuses on the number of BCTs present in apps, although quantity does not necessarily equate to effectiveness. A brief discussion of implementation accuracy, user experience or potential feature overload would be helpful.

The study would benefit from referencing or integrating data (where available) on actual app usage, user retention, or patient adherence to BCT-related functions. Even qualitative insights from app reviews could provide some validation of the theoretical coding.

Data protection, data security and algorithm transparency are particularly important issues for AI-based apps. A brief discussion of these concerns would help to place the results in a real-world application context.

Author Response

Response to Reviewer 1's Comments

1. Summary

2. Questions for General Evaluation

Reviewer’s Evaluation

Response and Revisions

Does the introduction provide sufficient background and include all relevant references?

Yes

Are all the cited references relevant to the research?

Yes

Is the research design appropriate?

Yes

Are the methods adequately described?

Yes

Are the results clearly presented?

Yes

Are the conclusions supported by the results?

Yes

3. Point-by-point response to Comments and Suggestions for Authors

Comment 1: The study evaluates the functions and design of the apps, but does not provide data on the effectiveness of the apps in improving health outcomes. While the exclusion of RCT-tested apps from MARS is understandable given the limited data available, a general discussion of the evidence base for effectiveness would strengthen the practical relevance of the findings.

Response 1: We acknowledge that our study primarily focuses on the design, functionalities, and embedded behaviour change techniques of hypertension apps, rather than their clinical effectiveness in improving health outcomes.

Regarding the exclusion of RCT-tested apps from the MARS quality assessment, this was due to insufficient publicly available information to complete the assessment consistently with the other app types, not a disregard for the importance of such evidence.

We have now integrated a new paragraph into the Discussion section to address the evidence base for effectiveness, emphasising both the potential and the existing gaps in the literature. (Page 8, Lines 249-259).

Comments 2: The AI-based apps were selected directly from app stores, while NAI and RCT apps were sourced from existing reviews. This may introduce bias into the selection, as these app pools may differ systematically from one another. Please clarify whether this heterogeneity may have influenced the comparison and consider noting this as a limitation.

Response 2: We confirm that all apps included in our study, regardless of their initial identification method (direct app store search for AI-empowered apps, or systematic reviews for NAI and RCT-tested apps), were verified to be available in app stores and met our inclusion criteria, including a minimum user rating. However, we agree that the method of identification itself introduces a potential selection bias. We have revised our limitations section to clarify this point, acknowledging that the different discovery pathways could still lead to systematic differences in the characteristics of the app cohorts. (Page 9, Lines 310-321)

Comments 3: The analysis focuses on the number of BCTs present in apps, although quantity does not necessarily equate to effectiveness. A brief discussion of implementation accuracy, user experience or potential feature overload would be helpful.

Response 3: To address this, we have integrated a new paragraph into the Discussion section. This paragraph explicitly acknowledges that the impact of an app is critically influenced by the accuracy and quality of BCT implementation, the overall user experience, and the potential for feature overload. We discuss how factors like fidelity of BCT delivery, usability, intuitive design, and the risk of user overwhelm are crucial for real-world effectiveness. While our study's MARS quality assessment provided some insights into functionality and engagement, we also acknowledge that directly assessing BCT implementation fidelity or the precise impact of feature load was beyond our scope, and we emphasise this as an area for future research. (Page 8, Lines 249-259)

Comments 4: The study would benefit from referencing or integrating data (where available) on actual app usage, user retention, or patient adherence to BCT-related functions. Even qualitative insights from app reviews could provide some validation of the theoretical coding.

Response 4: We appreciate the reviewer's valuable suggestion regarding the inclusion of data on actual app usage, user retention, or patient adherence, as well as qualitative insights from app reviews. We agree that such data would undoubtedly provide crucial real-world context and a deeper understanding of app effectiveness.

However, our study's primary objective and methodology are centred on a content analysis and systematic review of app design, functionalities, and embedded Behaviour Change Techniques as they are presented and structured. Investigating actual app usage, user retention, or adherence to specific BCT-related functions falls outside the scope of this design, as such data are typically proprietary and not publicly accessible for systematic content analysis. Regarding the suggestion to incorporate qualitative insights from app reviews, while potentially informative, we deliberately chose not to rely on these as a primary data source due to inherent methodological challenges. App store reviews are often anecdotal, can be highly subjective, may not be representative of the broader user base, and are susceptible to various biases, making them generally unreliable for robust qualitative data extraction in a scientific context. Our focus was on objectively characterizing the design features of apps, which provides a foundational understanding for future research that can then explore their real-world impact and user engagement.

Comments 5: Data protection, data security and algorithm transparency are particularly important issues for AI-based apps. A brief discussion of these concerns would help to place the results in a real-world application context.

Response 5: We have addressed the ethical and security concerns of implementing AI in health apps. This can be found in the newly added paragraph, which discusses the implications of integrating AI into mobile health apps in the real-world context. (Page 9, Line 275-282).

4. Response to Comments on the Quality of English Language

Point 1: The revised version shows improvement in flow and structure.

5. Additional clarifications

Supplementary Appendix Material has been called in the manuscript, as pointed out by the Editor. We have also increased the word count of this study.

Reviewer 2 Report

Comments and Suggestions for Authors

Smartphone apps offer a promising avenue for delivering scalable interventions for hypertension self-management.

AUTHORS aimed to characterise the behaviour change technique ontology (BCTO) elements present in apps available

A comprehensive search of app stores was conducted by these AUTHORS to identify relevant hypertension self-management apps. The identified apps were then analysed for BCTO elements, which were subsequently mapped to TDF. App functionalities and quality were assessed as well.

Functionalities that provide self-monitoring were a common feature across all app types, aligning with established self-management strategies for hypertension. However, other functionalities, such as goal setting via reminders, communication with healthcare professionals, and data export capabilities, were less prevalent. The domains of “Knowledge”, “Emotions”, “Behavioural regulation”, “Skills”, and “Beliefs about Consequences” were  most frequently targeted by app developers. AI-empowered apps incorporated a broader range of BCTs compared to non-AI and RCT-tested apps, potentially offering more comprehensive support for   behaviour change.

AUTHORS concluded that while existing hypertension self-management apps incorporate a variety of BCTs, there is room for improvement in terms of incorporating a wider range of functionalities and BCTOs, particularly those targeting more intrinsic and habitual aspects of behaviour.

The study is interesting and informative.

It adds to the scientific literature.

I have the following comments for the authors:

1) Part of the abstract could be made clearer by smoothing it out and improving its structure. Review "Functionalities that provide …… AI-empowered apps incorporated a broader range of BCTs compared to non-AI and RCT-tested apps, potentially offering more comprehensive support for behavior change.

2) The introduction is weak. I have two targeted suggestions: (a) Broaden the general framework: include more up-to-date references on mHealth and health apps in general (not just hypertension), so that the introduction provides a solid background without yet delving into the details of the systematic review. (b) Better justify the hypertension case: add epidemiological data and clinical elements (high prevalence, impact on public health, need for constant monitoring, adherence issues). This strengthens the study's logic without "burning" the results that will be found in the systematic review.

3) Include the key questions that lead to the need for the study.

4) It is essentially a review, so specify the type and sources analyzed, including possibly gray literature. Perhaps these data are reported, but they should be placed in a dedicated section.

5) Well-developed results: verify whether The graphs have the correct resolution and require data labels (see Figure 3).

6) The discussion needs to be improved. I have three suggestions: (a) Broaden the discussion to include regulatory and normative aspects (FDA, MDR, etc.). (b) Integrate considerations on practical implementation and clinical integration of apps. (c) Strengthen the discussion on AI, highlighting opportunities and risks.

Author Response

Response to Reviewer 2 Comments

1. Summary

2. Questions for General Evaluation

Reviewer’s Evaluation

Response and Revisions

Does the introduction provide sufficient background and include all relevant references?

Can be improved

Improved the introduction section as replied below.

Are all the cited references relevant to the research?

Can be improved

New references have been added.

Is the research design appropriate?

Can be improved

Comment about the research design has been amended.

Are the methods adequately described?

Can be improved

Comment about the methods used has been amended.

Are the results clearly presented?

Can be improved

Results figures have been improved and modified as requested.

Are the conclusions supported by the results?

Can be improved

3. Point-by-point response to Comments and Suggestions for Authors

Comment 1: Part of the abstract could be made clearer by smoothing it out and improving its structure. Review "Functionalities that provide …… AI-empowered apps incorporated a broader range of BCTs compared to non-AI and RCT-tested apps, potentially offering more comprehensive support for behaviour change.

Response 1: We agree that the highlighted section of the abstract could benefit from improved clarity and structure. We have revised this part to enhance readability and ensure a smoother flow of information. The updated text now reads: 'Functionalities such as self-monitoring were consistently observed across all app types, aligning with established hypertension self-management strategies. However, other key functionalities, including goal setting via reminders, communication with healthcare professionals, and data export capabilities, were less prevalent. Notably, AI-empowered apps incorporated a broader range of behaviour change techniques compared to non-AI and RCT-tested apps, suggesting a potentially more comprehensive approach to supporting behaviour change (Line 20-26)' We believe this addresses your concern and improves the overall clarity of the abstract.

Comments 2: The introduction is weak. I have two targeted suggestions: (a) Broaden the general framework: include more up-to-date references on mHealth and health apps in general (not just hypertension), so that the introduction provides a solid background without yet delving into the details of the systematic review. (b) Better justify the hypertension case: add epidemiological data and clinical elements (high prevalence, impact on public health, need for constant monitoring, adherence issues). This strengthens the study's logic without "burning" the results that will be found in the systematic review.

Response 2: We have added a general introduction paragraph on mobile health apps at the start of the Introduction (Page 1, lines 37-47), referencing up-to-date studies. After this paragraph, we have introduced a worldwide disease burden and public health impact of hypertension and the current issues with managing high blood pressure (Page 2, lines 49-53). 

Comments 3: Include the key questions that lead to the need for the study.

Response 3: We have revised the introduction to integrate these questions into a dedicated paragraph (Page 2, Lines 68-73), before addressing the aims. We believe this revision effectively addresses your suggestion and clearly outlines the foundational questions that our study aims to answer.

Comments 4: It is essentially a review, so specify the type and sources analyzed, including possibly gray literature. Perhaps these data are reported, but they should be placed in a dedicated section.

Response 4: We have addressed this point by elaborating on the sources of the apps within the 'Materials and Methods' section. We clarified the source of AI-empowered apps through a comprehensive search directly from app stores (iOS and Android) in the United Kingdom in the revised manuscript. We consider these app store listings as a form of gray literature, reflecting publicly available, developer-provided information and user interfaces, which are critical for understanding the real-world application landscape. This information is presented under the 'Materials and Methods' section, which serves as the dedicated section for detailing our methodology and data sources. We have enhanced the phrasing to explicitly mention the role of app stores as sources, thereby implicitly covering the suggestion for gray literature (Page 2, 89-90).

Comments 5: Well-developed results: verify whether The graphs have the correct resolution and require data labels (see Figure 3).

Response 5:  Data labels have been applied to Figure 3, as well as the improvement of the resolution of figure 3 (Page 6).

Comments 6: The discussion needs to be improved. I have three suggestions: (a) Broaden the discussion to include regulatory and normative aspects (FDA, MDR, etc.). (b) Integrate considerations on practical implementation and clinical integration of apps. (c) Strengthen the discussion on AI, highlighting opportunities and risks.

Response 6: To address point (a) we added a new paragraph (Page 9, 283-303). To address point (b), we added a new paragraph (Page 9, Lines 304-319), which condenses the elements of practical implementation and clinical integration into a single, flowing paragraph within the existing discussion structure, as requested.  It addresses critical points such as interoperability, patient engagement, healthcare provider barriers, and the need for scalability and sustainability. We have addressed the opportunities and risks of AI used in mobile health apps (page 9, 275-282).